# Association between altered cognition and *Loa loa* microfilaremia: First evidence from a cross-sectional study in a rural area of the Republic of Congo

**Thomas Checkouri[1], François Missamou[2], Sebastien D. S. Pion[3], Paul Bikita[2], Marlhand C. Hemilembolo[2,3], Michel Boussinesq[3], Cédric B. Chesnais[3]°, Jérémy T. Campillo [3]°***

**1** AP-HP, Service des Urgences Cérébro-Vasculaires, Hôpital Pitié-Salpêtrière, Sorbonne Université, Paris, France, **2** Programme National de Lutte contre l'Onchocercose, Direction de l'Épidémiologie et de la Lutte contre la Maladie, Ministère de la Santé et de la Population, Brazzaville, Republic of the Congo, **3** UMI 233 TransVIHMI, Université Montpellier, Institut de Recherche pour le Développement (IRD), INSERM Unité, Montpellier, France

☯ These authors contributed equally to this work.

* jeremy.campillo@ird.fr

**Data Availability Statement:** The authors confirm that all data underlying the findings are fully

## Abstract

### Background

Individuals with high *Loa loa* microfilarial densities are at risk of developing severe encephalopathy after administration of antiparasitic drugs. Apart from this finding, loiasis is considered benign with no effect on brain function. However, recent epidemiological data suggest an increased mortality and morbidity in *L. loa* infected individuals, underscoring the importance of studies on the possible neurological morbidity associated with loiasis.

### Methodology

Using MoCA tests and neurological ultrasounds, we conducted a cross-sectional study to assess cognitive alteration in a population living in a rural area endemic for loiasis in the Republic of Congo. Fifty individuals with high microfilarial densities (MFD) were matched on sex, age and residency with 50 individuals with low MFD and 50 amicrofilaremic subjects.

Analyses focused on individuals with MoCA scores indicating an altered cognition (i.e. < 23/30) and on the total MoCA score according to *Loa loa* MFD, sociodemographic characteristics and neurological ultrasound results.

### Principal findings

MoCA scores were very low in the studied population (mean of 15.6/30). Individuals with more than 15,000 microfilariae per milliliter of blood (mean predicted score:14.0/30) are more than twenty times more likely to have an altered cognition, compared to individuals with no microfilaremia (mean predicted score: 16.3/30). Years of schooling were strongly associated with better MoCA results. Extracranial and intracranial atheroma were not associated with *L. loa* MFD.

available without restriction. All relevant data are within the paper and its Supporting information file.

**Funding:** This study was fully supported by the European Research Council (ERC) under the European Union's Horizon 2020 research and innovation program (grant agreement No 949963, grant recipient: CBC). The funders had no role in the study design, data collection, data analysis, data interpretation, or writing of the report.

**Competing interests:** The authors have declared that no competing interests exist.

## Conclusion/significance

Loaisis microfilaremia is probably involved in cognitive impairment, especially when the MFD are high. These results highlight the urgent need to better understand loaisis-induced morbidity. Further studies investigating neurological morbidity of loiasis are needed.

### Author summary

Individuals with high *Loa loa* blood microfilarial densities are at risk of developing a serious encephalopathy after administration of antiparasitic drugs. Apart from this major inconvenience, *L. loa* filariasis (loiasis) is considered a benign disease, with no effect on brain function. However, recent epidemiological data suggest an increased mortality and morbidity in *L. loa* infected individuals, underscoring the importance of studies on the possible neurological morbidity associated with loiasis. This cross-sectional study assessed cognition using MoCA tests (Montreal Cognitive assessments, a tool for early detection of mild cognitive impairment) and neurological ultrasound among 143 matched individuals with either no microfilaremia, low microfilaremia or high microfilaremia. People with high microfilaremia were more than twenty times more likely to have an altered cognition, compared to individuals with no microfilaremia.

## Introduction

Loiasis is a parasitic disease caused by the filarial nematode *Loa loa*. It is transmitted from human to human by tabanids (*Chrysops* spp.), mainly in forested areas of Central Africa. Female worms can produce millions of larvae ("microfilariae", mf) that circulate in the bloodstream with a characteristic diurnal periodicity. The most frequent manifestations of loiasis are episodes of pruritus, sub-conjunctival migrations of the adult worms ("African eyeworm"), and transient angioedema ("Calabar swellings"). It is estimated that more than 15 million people are infected with *L. loa* in the world [1]. As of 2023, the disease is still considered benign: it is not included in the World Health Organization's list of Neglected Tropical Diseases, and thus does not benefit from control campaigns. This dogma has been recently questioned [2] after specifically designed population studies found an increased mortality in *L. loa*-infected individuals [3].

   Neurological attention has been drawn towards loiasis since cases of acute, fatal, encephalopathies were reported in infected individuals after treatment with ivermectin, hampering mass treatment for onchocerciasis ("river blindness") in loiasis coendemic areas [4]. The main risk factor for Ivermectin-associated encephalopathy (IAE) is to have high levels of *L. loa* microfilaremia ($> 30,000$ mf per milliliter of blood [mf/mL]). The causative mechanism of IAE is not fully known, but pathological findings are consistent with a rupture of the brain-blood barrier (mf found in the cerebrospinal fluid) and hemorrhages due to an accumulation of mf in the brain microcirculation [5]. Besides this acute drug-related adverse event, various case reports have associated untreated loiasis with neurological disorders: spontaneous encephalopathies, severe headaches refractory to medical treatment, or nerve palsies [6]. Despite these alarming data about potential neurological consequences of loiasis, no epidemiological studies have ever been conducted to evaluate the brain function of *L. loa*-infected patients.

The Montreal Cognitive Assessment (MoCA) score is a widely used screening tool for mild cognitive impairment (MCI) and dementia, addressing numerous cognitive domains [7]. The score ranges from 0 to 30, with higher scores indicating better cognition. The diagnosis of MCI requires evidence of impairment in at least one of those cognitive domains, including executive function, attention, memory, language or visuospatial skills, associated with cognitive complaint from the patient or observant [8]. There are many other recommended neuropsychological tests to confirm alteration of cognition, such as Mini-cog or Dementia severity rating scale (DSRS) [9]. The diagnosis of dementia is done when the cognitive impairment is sufficient to interfere with the complex activities of daily living [10]. Unlike other screening tools, MoCA score has been translated and used in multiple African countries, including in low-resource settings [11], and with illiterate patients [12]. Initially developed for neurodegenerative diseases such as Alzheimer's in high income resources settings, it also has been used to screen for MCI and dementia in infectious (HIV-associated neurocognitive disorders) [13], genetic (Huntington's disease) [14], or vascular (post-stroke dementia) [15] diseases. The original cut–off score of 26/30 was validated in high-income countries with wide access to education, but a 2017 meta-analysis suggested a better classification accuracy with a lower cut-off at 23/30, especially in more culturally and educationally heterogenous settings [16]. To address cognitive alteration in specific populations with no available normative values, the use of MoCA as a continuous score with regression-based corrections accounting for confounding factors has also been suggested [17,18].

Other highly prevalent infectious diseases in sub-Saharan Africa have been associated with MCI and progression to dementia, such as cerebral malaria [19] and onchocerciasis [20]. Similar to high-income countries, cerebrovascular diseases with large artery atheroma and cerebral microangiopathy are major contributors to the burden of dementia [21,22]. Strict adjustment for these confounding factors has been limited due to the lack of neurological specialists and access to brain magnetic resonance imaging (MRI) in rural areas [23].

Addressing the growing concerns about the understated neurological morbidity in loiasis, this study aims to assess whether cognitive alteration can be associated with *L. loa* infection, and its relation with *L. loa* microfilarial density levels.

## Methods

### Ethics statement

This study has been approved by the Ethics Committee of the Congolese Foundation for Medical Research (N° 036/CIE/FCRM/2022) and by the Congolese Ministry of Health and Population (N° 376/MSP/CAB/UCPP-21). All participants received clear and appropriate information and signed an informed consent form for this specific study.

### Study area and population

The MorLo (*Morbidity due to Loiasis*) project is an international collaborative study to assess the prevalence and incidence of *Loa*-related organ-specific complications in rural sub-Saharan Africa where loiasis is endemic and hypoendemic for onchocerciasis. In 2022, a cohort including 991 individuals living in 21 villages located in a radius of 50 kilometers around Sibiti, the capital town of the Lékoumou district of the Republic of Congo, was initiated. This region was selected because no mass drug administration with ivermectin for onchocerciasis or lymphatic filariasis had ever been implemented. To investigate neurological morbidity, we conducted this pilot cross-sectional study nested in the cohort study. Three groups were constituted within the whole cohort, to ensure adequate comparability. Each individual with high *L. loa* microfilarial density (MFD) (> 8000 mf/mL) was matched on sex, age (± 5 years) and village

of residence with two others: one subject with low *L. loa* MFD (500 to 7999 mf/mL) and one amicrofilaremic individual.

## Patient and public involvement statement

This study is nested in the cohort set up within the Sibiti hospital as part of the MorLo project. The project protocol was developed in collaboration with community health workers in the villages involved. Nurses from the Sibiti hospital administered the neurocognition tests after training by a neurologist. Community health workers acted as communicators within the involved villages to encourage people to visit health facilities when they had infectious symptoms as part of the MorLo cohort. The main results of the study were provided to community health workers in the involved villages. For each patient, the results of their tests were safely and confidentially kept at the health facility of their village and were accessible upon request.

## Laboratory procedures

In May 2022, 50 μL of blood was collected by finger prick from each participant and spread on a microscope slide to prepare thick blood smear (TBS) between 10 am and 4 pm to account for the *L. loa* microfilaremia diurnal periodicity. The slides were dried at ambient temperature, dehemoglobinized and stained with Giemsa within 4 hours. All TBS were examined under a microscope at 100× magnification by experimented technicians to count the *L. loa* mfs and *Mansonella perstans* (another filariasis) mfs. Each TBS was read twice and the arithmetic mean of the counts was used for the statistical analysis.

Finally, from blood collected in a heparinized tube, a *L. loa* Antibody Rapid Test (*Loa* ART, Drugs & Diagnostics for Tropical Diseases, San Diego, California) and an *Ov*16 Rapid Diagnostic Test (*Ov*16 RDT; Drugs & Diagnostics for Tropical Diseases, San Diego, California) were performed for each patient and read with a HRDR-200 smartphone-based chromatographic reader, as recommended by the manufacturer. Two skin snips were collected from each patient with positive *Ov*16 RDT using a Holth-type corneoscleral punch and incubated in saline at room temperature for 24 hours. Emerged microfilariae (mf) were counted using a microscope, and the individuals' microfilarial density (MFD), expressed as mf per snip, were calculated using the arithmetic mean of the two snips.

## MoCA testing

As a preparatory phase to the study, two local nurses were trained to MoCA testing, under the supervision of a neurologist, with healthy subjects. Then, during the study, the two nurses conducted the MoCA testing in a quiet individual room, after explaining the purpose of the test. Nurses were blinded to the parasitological results. Patients could answer the questions in French, Lingala or Kikongo (local languages), at their convenience. The number of years of education was self-reported. For the denomination task, we selected the lion, the crocodile and the snake from the 3 validated versions. Seven sub-scores were calculated: executive function (alternating trail making and verbal fluency), visuospatial abilities (figure copy and clock drawing), attention, concentration and working memory (digit span, sustained attention and serial subtraction), language (animal naming and sentence repetition), abstract reasoning (word and concept association), orientation (giving the calendar date and the location) and memory (restituting the list of 5 words given in French, Lingala or Kikongo as asked by the patient). Patients with MoCA scores lower than 23/30 were considered to present an altered cognition (AC). All MoCA results were rechecked by a neurologist (TC) blinded to the parasitological results.

## Neurological ultrasounds

To examine a potential confounding factor for MCI, all patients had a cervical and transcranial Doppler and ultrasound examination with a CX–50 echograph (Phillips) and the appropriate probes. Cervical atheroma was defined as an intima-media thickness > 1.5 mm or presence of a plaque (focal structure encroaching at least 0.5 mm in the lumen) according to the Mannheim consensus [24]. We also measured blood velocities to characterize the degree of cervical stenosis, using the NASCET criteria [25]. Transcranial Doppler examination was done via the temporal and suboccipital bone windows, and we measured maximal systolic velocities (MSV), end diastolic velocities (EDV) and mean velocities (MV) from the middle cerebral artery (MCA). The pulsatility index (PI) was calculated automatically by the equation [(MSV-EDV)/MV]. Intracranial large artery atheroma was defined as blood velocities > 220/155 cm/s for MCA and > 140/100 cm/s for the basilar artery (BA), as published [26]. Considering PI as a surrogate marker of possible cerebral small-vessel disease, we defined a PI value of > 1.1 as a marker of microangiopathy, adapted from published criteria [27]. The missing data regarding the variables issued from the PI are due to a lack of temporal bone window. All ultrasound examinations were performed blinded to the MoCA and parasitological results.

## Statistical analysis

Variables of interest were: the presence of AC (MoCA score < 23/30) and the MoCA score (continuous variable). Explanatory variables of interest were: sex, age (categorized as following: 18–40, 41–50, 51–60 and > 60 years old), number of years of schooling (continuous variable), history of cerebral malaria, high blood pressure (defined by a systolic pressure higher than 140 mmHg and/or a diastolic pressure higher than 90 mmHg; yes/no), smoking status, presence of *L. loa* mf in the TBS (microfilaremic status), *Loa* ART result and *L. loa* MFD (as a continuous variable, and in balanced categories: 0, 1–1999, 2000–6999, 7000–14,999 and ≥ 15,000 mf/mL), *Ov*16 results, presence of large artery atheroma and markers of cerebral microangiopathy.

Ultrasound examination and cognitive evaluation results of participants were described and compared according to microfilaremic status. Chi-2 test (with Yates correction if one of the expected counts was lower than 5) and Kruskal-Wallis rank test were performed for categorical and continuous variables, respectively.

Bivariate and multivariable logistic regression models were used to assess the associations between MoCA results and explanatory variables. Probabilities to have an AC (MoCA score < 23) according to *L. loa* MFD were calculated from the multivariable saturated logistic regression model. Using Poisson regression, MoCA score was analyzed as a continuous score to address considerable variation compared to the normative MoCA values developed in high income countries. Indeed, the lack of previously published normative values in similar settings (rural population of Central Africa), the relatively small sample size and the non-normal distribution of MoCA scores did not allow the use of Z–scores derived from the group with no microfilaremia. An analysis of each sub-item was performed with the appropriate models for the variables to be explained (negative binomial, Poisson or ordinal regression) and presented in S1 Table. Finally, we performed sensitivity analyses using different cut-offs (< 20, 21 and 22/30) for the definition of AC (S2 Table).

The data used for analysis and the STROBE checklist for cross-sectional studies are available in S1 Data and S1 Strobe Checklist, respectively.

## Results

Of the 150 individuals invited to participate, 143 were included in the study (3 were absent because they went hunting, 1 was away from the village and 3 were excluded because of hearing or visual deficiency). Demographics, cognitive and ultrasound data of the 3 groups are shown in Table 1. Only 2 and 9 subjects had significant (> 50% stenosis) extracranial and intracranial atheroma, respectively. A total of 8 individuals had markers of cerebral microangiopathy. No difference was found according to microfilaremic status. A total of 26 subjects (18.2%), of whom 80.8% were females, had no temporal bone window. For these individuals, no PI value was measured and intracranial atheroma was assessed in the BA.

Only two individuals had mfs of *M. perstans* in their blood, both in low densities (60 and 600 mf/mL). Finally, only 8 individuals tested positive for *Ov*16. Of the 8, none had *O. volvulus* mfs in the skin.

Characteristics of the study population according to the presence of an AC are presented in Table 2. Females were more represented in the cognitive alteration category ($P$ = 0.014), and a higher number of years of education was strongly associated with no cognitive alteration ($P <$ .001). In the population, arithmetic mean MoCA score was 15.6 (range: 3–28). The least

**Table 1. Neurological examination and MoCA results according to loiasis status.**

| Characteristic | Whole population | Non-microfilaremic group | Microfilaremic group | | | |
| --- | --- | --- | --- | --- | --- | --- |
| | | | All | 500–7999 mf/mL | ≥ 8000 mf/mL | *P* * |
| Number of patients (n, %) | 143 | 47 (32.9%) | 96 (67.1%) | 47 | 47 | |
| **Ultrasound examination** | | | | | | |
| Extracranial atheroma | | | | | | |
| Carotid IMT [mm] | | | | | | .753 |
| Mean ± SD | 0.67 ± 0.18 | 0.66 ± 0.17 | 0.67 ± 0.19 | 0.69 ± 0.19 | 0.65 ± 0.19 | |
| Median [IQR] | 0.63 [0.53–0.78] | 0.61 [0.53–0.76] | 0.64 [0.53–0.80] | 0.69 [0.53–0.77] | 0.60 [0.51–0.80] | |
| Any ICA atheroma | 20 (14.1%) | 7 (14.9%) | 13 (13.7%) | 9 (19.1%) | 4 (8.5%) | .876 |
| ICA atheroma > 50% | 2 (1.4%) | 1 (2.1%) | 1 (1.1%) | 1 (2.1%) | 0 | NC |
| Intracranial atheroma | | | | | | |
| MCA stenosis > 50% | 9 (6.3%) | 2 (4.3%) | 7 (7.4%) | 4 (8.5%) | 3 (6.4%) | .474 |
| Markers of microangiopathy | 8 (5.6%) | 3 (6.4%) | 5 (5.2%) | 2 (4.3%) | 3 (6.3%) | .261 |
| **Cognitive evaluation** | | | | | | |
| Total MoCA score | | | | | | .454 |
| Mean ± SD | 15.6 ± 5.8 | 16.2 ± 5.7 | 15.3 ± 5.9 | 15.5 ± 6.2 | 15.1 ± 5.7 | |
| Median [IQR] | 16 [11–20] | 16 [11–20] | 15.5 [11–19] | 15 [11–20] | 15 [10–19] | |
| Sub–scores (Mean ± SD) | | | | | | |
| Executive function (/2) | 0.30 ± 0.57 | 0.42 ± 0.65 | 0.24 ± 0.52 | 0.32 ± 0.59 | 0.17 ± 0.43 | .061 |
| Visuospatial ability (/4) | 1.38 ± 1.44 | 1.19 ± 1.36 | 1.48 ± 1.48 | 1.55 ± 1.50 | 1.38 ± 1.47 | .299 |
| Attention (/6) | 2.33 ± 1.96 | 2.66 ± 1.92 | 2.18 ±1.97 | 2.51 ± 1.99 | 1.85 ± 1.94 | .142 |
| Language (/5) | 3.82 ± 1.20 | 3.77 ± 1.22 | 3.84 ± 1.19 | 3.62 ± 1.41 | 4.04 ± 0.90 | .714 |
| Abstract reasoning (/2) | 1.63 ± 0.63 | 1.68 ± 0.51 | 1.60 ± 0.69 | 1.47 ± 0.75 | 1.72 ± 0.61 | .888 |
| Memory (/5) | 1.27 ± 1.77 | 1.47 ± 1.80 | 1.17 ± 1.76 | 1.38 ± 1.82 | 0.98 ± 1.69 | .282 |
| Orientation (/6) | 4.91 ± 1.33 | 5.06 ± 1.17 | 4.83 ± 1.40 | 4.7 ± 1.56 | 4.98 ± 1.23 | .430 |

Abbreviations: SD, Standard deviation; IQR, interquartile range; NC, not calculable; IMT, Intima media thickness; ICA, Internal carotid artery; MCA, Middle cerebral artery

* represents the p-value associated with the Chi-2 or Kruskal-Wallis rank test (according to the variable) between the microfilaremic and non-microfilaremic groups.

**Table 2. Characteristics of the study population according to altered cognition status.**

| Characteristic | Whole population | Altered Cognition* | No Altered Cognition | *P* |
|---|---|---|---|---|
| Number of patients (n, %) | 143 | 122 (85.3%) | 21 (14.7%) | |
| Age in years (mean ± SD) | 51.4 ± 13.9 | 51.9 ± 13.5 | 48.4 ± 15.9 | .321 |
| Sex-ratio (M/F) | 2.0 | 1.7 | 9.5 | .014 |
| Hypertension (n, %) | 40 (28.0%) | 36 (29.5%) | 4 (19.1%) | .324 |
| Tobacco use (n, %) | 33 (23.1%) | 28 (23.0%) | 5 (23.8%) | .931 |
| Cerebral malaria history (n, %) | 12 (8.4%) | 1 (4.8%) | 11 (9.0%) | .516 |
| Years of education (mean ± SD) | 4.8 ± 3.9 | 4.0 ± 3.6 | 9.3 ± 2.3 | < .001 |
| No school education (n, %) | 39 (27.3%) | 39 (32.0%) | 0 | |
| 1–6 years education (n, %) | 55 (38.5%) | 52 (42.6%) | 3 (14.3%) | |
| > 6 years education (n, %) | 49 (34.3%) | 31 (25.4%) | 18 (85.7%) | |
| *Ov16* Rapid Diagnostic Test (n, %) | | | | .857 |
| Positive | 8 (5.6%) | 7 (5.7%) | 1 (4.8%) | |
| Negative | 135 (94.4%) | 20 (95.2%) | 115 (94.5%) | |
| *Loa* Antibody Rapid Test (n, %) | | | | .187 |
| Positive | 117 (86.0%) | 98 (84.5%) | 19 (95.0%) | |
| Negative | 19 (14.0%) | 18 (15.5%) | 1 (5.0%) | |
| MD | 7 | 6 | 1 | |
| *Loa* microfilaremic status (n, %) | | | | .291 |
| Positive | 96 (67.1%) | 84 (68.9%) | 12 (57.1%) | |
| Negative | 47 (32.9%) | 38 (31.1%) | 9 (42.9%) | |
| *Loa* MFD (mf/mL) | | | | .159 |
| Mean ± SD | 7783 ± 12 209 | 8482 ± 12 944 | 3723 ± 4943 | |
| Median [IQR] | 2220 [0–12,240] | 3145 [0–12,610] | 1770 [0–4450] | |
| *Loa* MFD categories (n, %) | | | | .380 |
| 0 mf/mL | 47 (32.9%) | 38 (31.1%) | 9 (42.9%) | |
| 1–4999 mf/mL | 44 (30.8%) | 37 (30.3%) | 7 (33.3%) | |
| 5000–14 999 mf/mL | 27 (18.9%) | 23 (18.9%) | 4 (19.0%) | |
| ≥15 000 mf/mL | 25 (17.5%) | 24 (19.7%) | 1 (4.8%) | |

Abbreviations: SD, Standard deviation; MFD, microfilarial density; IQR, interquartile range.

* defined if MoCA total score < 23

successful parts were executive function (0.30/2) and memory (1.27/5) and the most successful ones were orientation (4.91/6) and abstract reasoning (1.63/2). Verbal fluency was below than 11 words in 91.6% of cases, with an average number of 4.3 words given.

Results from the logistic regression model analyses explaining the presence of AC are presented in Table 3. Individuals with ≥ 15,000 mf/mL were 21.80 [95% CI: 1.25–380.15] times more likely to have an AC, compared to individuals with no microfilaremia (*P* = .043). Probabilities of having an AC, calculated from this saturated model, were estimated at 0.80 and 0.97 among individuals with 0 and ≥ 15,000 mf/mL, respectively. Predicted probabilities of having an AC (calculated according to the number of years of schooling and the MFD categories) highlighted a significantly higher risk of having an AC when the MFD is high (≥ 15,000 mf/ mL), particularly for individuals with the highest education levels (Fig 1, panel A). Sensitivity analyses using different MoCA cut-offs were consistent with the main analysis results, with a significant (or borderline significant) association found with high *Loa* MFD levels for each cut-off (S2 Table).

**Table 3. Results from the logistic regression models explaining altered cognition status.**

| Variable | cOR [95% CI] | P | aOR [95% CI] | P |
|---|---|---|---|---|
| Age (years old) | | | | |
| 18–40 | Ref. | | Ref. | |
| 41–50 | 1.81 [0.52, 6.35] | .351 | 2.40 [0.34, 17.00] | .380 |
| 51–60 | 2.07 [0.48, 8.93] | .327 | 1.54 [0.15, 15.67] | .715 |
| > 60 | 1.56 [0.47, 5.16] | .470 | 0.20 [0.02, 2.05] | .174 |
| Sex | | | | |
| Female | Ref. | | Ref. | |
| Male | 0.18 [0.04, 0.81] | .025 | 0.38 [0.04, 3.64] | .401 |
| History of cerebral malaria (ref. No) | 1.98 [0.24, 16.21] | .524 | 0.75 [0.03, 18.38] | .860 |
| High blood pressure (ref. No) | 1.78 [0.56, 5.66] | .329 | 0.83 [0.11, 6.05] | .856 |
| Use of smoking tobacco (ref. No) | 0.95 [0.32, 2.83] | .931 | 0.41 [0.06, 2.59] | .342 |
| **Years of schooling (continuous)** | **0.62 [0.50, 0.76]** | **< .001** | **0.53 [0.40, 0.70]** | **< .001** |
| *L. loa* MFD (mf/mL) | | | | |
| 0 | Ref. | | Ref. | |
| 1–1999 | 1.50 [0.36, 6.19] | .575 | 3.00 [0.33, 27.37] | .330 |
| 2000–6,999 | 1.24 [0.34, 4.53] | .741 | 1.20 [0.15, 9.74] | .859 |
| 7000–14,999 | 1.18 [0.32, 4.33] | .798 | 1.55 [0.21, 11.19] | .663 |
| **≥ 15,000** | **5.68 [0.68, 47.75]** | **.110** | **21.80 [1.09, 433.05]** | **.043** |
| Loiasis Rapid Antibody Test | | | | |
| No | Ref. | | Ref. | |
| Yes | 0.29 [0.03, 2.28] | .237 | 0.12 [0.01, 2.81] | .189 |
| MD | 0.33 [0.02, 6.19] | .461 | 0.14 [0.01, 11.95] | .384 |
| Presence of large artery atheroma (ref. No) | 2.47 [0.54, 11.33] | .243 | 3.14 [0.37, 26.56] | .293 |
| Possible cerebral microangiopathy | | | | |
| No | Ref. | | Ref. | |
| Yes | 0.55 [0.10, 2.98] | .492 | 0.08 [0.01, 1.73] | .107 |
| MD | 2.22 [0.48, 10.27] | .308 | 0.19 [0.02, 2.22] | .188 |

Abbreviations: cOR, crude Odds-Ratio; aOR, adjusted Odds-Ratio; 95% CI, 95% confidence intervals; MD, Missing data

Results from the Poisson regression explaining total MoCA score are presented in Table 4. Individuals with MFD ≥ 15,000 mf/mL had a mean predicted MoCA score of 14.0 compared to 16.3 for amicrofilaremic individuals. Years of schooling were highly associated with better MoCA scores with a mean predicted score of 11.1 for individuals who have not been to school, 15.2 for those who have been to school for 5 years and 20.9 for those who have been to school for 10 years. Predicted MoCA scores (calculated according to the number of years of schooling and the MFD categories) showed significantly lower scores in the higher MFD category, for any level of education (Fig 1, panel B).

Mean MoCA total scores and each sub-item scores are presented by age, sex, *Loa* microfilaremic status, and MFD categories in S3 Table. High *L. loa* MFD was significantly associated with lower results in executive function ($P$ = .028, trend test of Cuzick) and attention ($P$ = .015, trend test of Cuzick) exercises. Results from models explaining each MoCA sub-items are presented in S1 Table. By comparison to amicrofilaremic subjects, individuals with ≥ 15,000 mf/mL were more likely to have poor results for memory (adjusted incidence rate ratio [aIRR] = 0.28; $P$ = .014), attention (aIRR = 0.55; $P$ = .003) and executive function (adjusted Odds Ratio = 0.16, $P$ = 0.036) exercises.

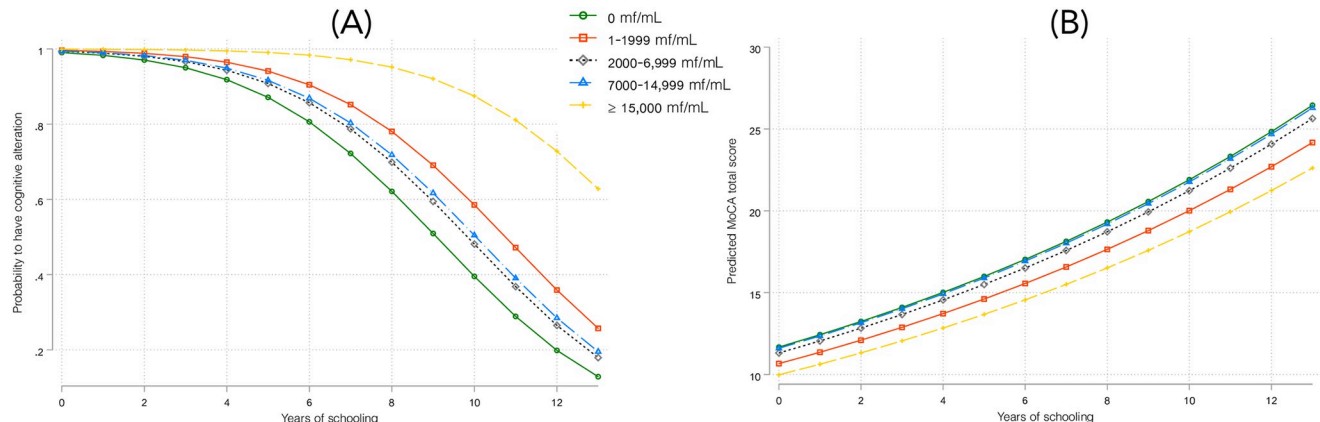

**Fig 1. Prediction of altered cognition and total MoCA score as affected by years of schooling and *Loa loa* microfilarial densities.** Panel A. Probabilities to have an altered cognition. Panel B. Total MoCA scores.

**Table 4. Results from the Poisson regression models explaining MoCA total score.**

| Variable | cCoeff. [95% CI] | P | aCoeff. [95% CI] | P |
|---|---|---|---|---|
| Age (years old) | | | | |
| 18–40 | Ref. | | Ref. | |
| 41–50 | -0.10 [-0.21, 0.01] | .075 | -0.10 [-0.22, 0.02] | .095 |
| 51–60 | -0.28 [-0.41, -0.15] | < .001 | -0.12 [-0.26, 0.02] | .088 |
| > 60 | -0.37 [-0.48, -0.25] | < .001 | -0.12 [-0.26, 0.03] | .119 |
| Sex | | | | |
| Female | Ref. | | Ref. | |
| Male | 0.34 [0.24, 0.43] | < .001 | 0.10 [-0.02, 0.22] | .121 |
| History of cerebral malaria (ref. No) | 0.01 [-0.14, 0.16] | .862 | 0.06 [-0.10, 0.22] | .454 |
| High blood pressure (ref. No) | -0.22 [-0.32, -0.12] | < .001 | 0.01 [-0.10, 0.12] | .870 |
| Use of smoking tobacco (ref. No) | 0.10 [0.00, 0.20] | .041 | 0.05 [-0.06, 0.16] | .350 |
| **Years of schooling (continuous)** | **0.07 [0.06, 0.08]** | **< .001** | **0.06 [0.05, 0.07]** | **< .001** |
| *L. loa* MFD (mf/mL) | | | | |
| 0 | Ref. | | Ref. | |
| 1–1999 | -0.01 [-0.14, 0.11] | .840 | -0.09 [-0.22, 0.04] | .184 |
| 2000–6,999 | -0.08 [-0.21, 0.04] | .188 | -0.03 [-0.16, 0.10] | .618 |
| 7000–14,999 | 0.04 [-0.08, 0.16] | .490 | 0.01 [-0.13, 0.13] | .980 |
| **≥ 15,000** | **-0.18 [-0.31, -0.05]** | **.005** | **-0.16 [-0.29, -0.02]** | **.022** |
| Loiasis Rapid Antibody Test | | | | |
| No | Ref. | | Ref. | |
| Yes | 0.13 [-0.01, 0.26] | .052 | 0.03 [-0.10, 0.17] | .563 |
| MD | 0.19 [-0.03, 0.41] | .087 | -0.01 [-0.25, 0.22] | .898 |
| Presence of large artery atheroma (ref. No) | -0.22 [-0.33, -0.10] | < .001 | -0.08 [-0.21, 0.04] | .182 |
| Possible cerebral microangiopathy | | | | |
| No | Ref. | | Ref. | |
| Yes | -0.27 [-0.47, -0.07] | .008 | -0.04 [-0.25, 0.16] | .669 |
| MD | -0.26 [-0.38, -0.15] | < .001 | 0.05 [-0.10, 0.20] | .507 |

Abbreviations: cCoeff, crude ß coefficients; aCoeff, adjusted ß coefficients; 95% CI, 95% confidence intervals; MD, Missing data

## Discussion

Using a cross-sectional design, we report an association between low MoCA scores and *L. loa* microfilaremia. To our knowledge, this is the first epidemiological study to report neurological morbidity associated with loiasis.

We used MoCA testing, a screening tool for MCI and dementia mainly used in high-income countries and not validated for the specific population investigated. The majority of patients had a positive screening for cognitive impairment (defined by a MoCA score < 23), probably related to the fact that normative values of MoCA are different in this population of rural Congo. However, previous studies had shown high sensibility and specificity (83% and 88%, respectively) for dementia using the 23/30 cut-off [16]. MoCA testing has been successfully used in rural areas in Africa to detect confirmed infectious causes of dementia such as HIV-associated neurocognitive disorders, even if the authors noted the limitation of accuracy due to differences in socio-cultural context [28,29]. Although the significant association between AC and loiasis was observed in patients with high MFD, we observed no clear dose–effect of *L. loa* MFD on MoCA score. This may reflect a threshold effect at which the neurocognitive effect would appear, or a lack of power due to sample size. Further studies will help to better understand this phenomenon and to evaluate the validity of this MoCA in rural areas of Central Africa. Nevertheless, the approach using continuous MoCA scores with strict adjustment for confounding socio-demographic and pathological factors, as well as the difference found mainly in patients having attended school for more than 10 years (that limits a potential floor effect), strengthens the likelihood of our results. Overall, we demonstrate a specific alteration of cognition in individuals with *L. loa* microfilaremia.

After screening with MoCA, a neuropsychological analysis of specifically altered cognitive domains is needed to confirm MCI. The MoCA sub-items of memory, attention, and executive function were selectively impacted by *L. loa* MFD in this study. All those subdomains of MoCA testing are recognized in international criteria as objective evidence to diagnose MCI, suggesting a genuine effect of *L. loa* MFD on cognition [9]. These results concerning MoCA sub-items should be interpreted with caution, as that they are strongly influenced by illiteracy status [12]. However, we adjusted for educational level and the sampled population was homogeneous (subjects lived in similar villages in terms of access to education and socio-demographic characteristics), strengthening the plausibility of the results. The further step to diagnose dementia in this population would be the assessment of alteration of the subjects' autonomy in daily living. In the specific context of the Republic of Congo, the situation is particularly concerning with social stigma and absence of caregivers reported for patients suffering from dementia [30]

The pathogenesis of neurocognitive dysfunction in the context of chronic microfilaremia is yet to discover. Knowledge about potential brain lesions due to *L. loa* mainly comes from cases of IAE. Patients present with a progressive comatose state with diffuse electroencephalographic slowing, acute proteinuria, and lumbar puncture revealing the presence of living mf in the cerebrospinal fluid [5]. An alteration of the retinal microcirculation, with "cotton wool exudates" and retinal hemorrhages were reported in cases of IAE, but also in untreated individuals with high *L. loa* MFD [31,32]. Disruption of the blood–brain barrier by dead microfilariae or proinflammatory cytokines has also been suggested as a mechanism in historical cases of spontaneous encephalopathies [33]. Emboli of dead mf in small cerebral vessels, surrounded by parenchymal granulomas were found during the autopsy of baboons experimentally infected with *L. loa* after treatment with ivermectin, but also in untreated animals [34]. In humans, perivascular granulomas were reported in a subject who died after having developed an IAE

[35]. Interestingly, a high MFD is the main risk factor for IAE [4], and only individuals with high MFD had an alteration of cognition in this study.

Considering a possible similarity between the mechanism of IAE and the alteration of cognition found in chronically infected individuals in our study, two main mechanisms are to be tested. A first possible mechanism is an impairment of cerebral microcirculation by the chronic circulation of mf. Cerebral microcirculation is also altered in cerebral small-vessel disease, which is a major cause of MCI and dementia in high income countries [36]. The use of PI as a surrogate marker for microangiopathy, as well as the proportion (18.2%) of patients without temporal bone window are major limitations on the interpretability of our results for this hypothesis. Other possible mechanisms are related to the presence of the parasite in the cerebrospinal fluid or in the subdural space [37–40], or disruption of the blood–brain barrier [33]. Even if lumbar puncture and brain imaging were not available, the absence of chronic headaches and visual impairment in our patients makes this mechanism unlikely. The relationships between *L. loa* MFD, cerebrovascular disease and cognitive impairment remain to be studied. Further investigation on the mechanism of cognitive alteration from loiasis would require brain MRI, which is not available in rural Congo, or lumbar puncture, an invasive procedure with no guaranteed benefit/risk ratio.

Other infections may explain the altered cognition in this population such as onchocerciasis and intestinal worms (hookworm whipworm and roundworm). Indeed, uncontrolled seizures could result from epilepsy associated with onchocerciasis [41]. Those seizures could alter cognition, independently of *L. loa* infection. However, only 8 individuals in the study population had a positive rapid serological test for *Ov*16 and of these, none had mfs in the skin. Intestinal worms, particularly hookworms, have been described as being involved in impairments in intellectual and cognitive development in children [42–44]. These are common conditions in the study population, but there is no reason why individuals with *L. loa* microfilaremia should carry more intestinal worms than individuals without. It is therefore unlikely that the presence of intestinal worms is a confounding factor in our study.

Our study has several limitations. The cross–sectional design is not sufficient to establish a causal relationship between *L. loa* infection and neurocognitive disease. The lack of normative values for the screening test used in our population, as well as the relatively small sample size clearly limits the interpretability of our results. These limitations are inherent to the "pilot-study" design aimed to identify potential neurological morbidity in a disease which is largely considered as benign. To address these limitations, larger studies with comprehensive and validated cognitive testing, long term follow–up, brain imaging and adjustment for coinfections are currently under planning.

Overall, the alteration of global cognition and critical cognitive domains by *L. loa* microfilaremia in clinically meaningful proportions could place loiasis as a significant risk factor for dementia in Central Africa, especially for the young population. This could add to the growing socio-economic burden of dementia in rural Africa, with the lack of efficient diagnostic procedures and care systems for these patients. Our results provide the rationale for further studies investigating the impact of *L. loa* microfilaremia on cognitive function. This should push further the consideration that loiasis is not a benign disease.

## Supporting information

**S1 Table. Sub-items models (binomial negative, Poisson or ordinal) with *L. loa* microfilarial densities in categories.**
(DOCX)

**S2 Table. Sensitivity analysis: Adjusted logistic regression explain using different MoCA score cut-offs for the definition of altered cognition.**
(DOCX)

**S3 Table. MoCA scores by age, sex, *Loa* microfilaremic status and *L. loa* MFD.**
(DOCX)

**S1 Strobe Checklist. STROBE Statement: Checklist of items that should be included in reports of cross-sectional studies.**
(DOCX)

**S1 Data. Anonymized data.**
(CSV)

## Acknowledgments

We are grateful to local authorities for their help and support. We thank the individuals in the study for agreeing to participate. We thank the health personnel of the Sibiti hospital for their participation in the study. We thank community health workers of each village for their help and advices. We thank the drivers for their work.

## Author Contributions

**Conceptualization:** Thomas Checkouri, Jérémy T. Campillo.

**Data curation:** Cédric B. Chesnais.

**Formal analysis:** Cédric B. Chesnais.

**Funding acquisition:** Cédric B. Chesnais.

**Investigation:** Thomas Checkouri, François Missamou, Paul Bikita, Marlhand C. Hemilembolo, Cédric B. Chesnais, Jérémy T. Campillo.

**Methodology:** Thomas Checkouri, Cédric B. Chesnais, Jérémy T. Campillo.

**Project administration:** Jérémy T. Campillo.

**Resources:** Jérémy T. Campillo.

**Supervision:** François Missamou, Jérémy T. Campillo.

**Validation:** Jérémy T. Campillo.

**Visualization:** Jérémy T. Campillo.

**Writing – original draft:** Thomas Checkouri.

**Writing – review & editing:** François Missamou, Sebastien D. S. Pion, Paul Bikita, Marlhand C. Hemilembolo, Michel Boussinesq, Cédric B. Chesnais, Jérémy T. Campillo.

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
