## [Decision Letter · Decision Letter 0]

15 May 2023

Dear Dr. Checkouri,

Thank you very much for submitting your manuscript "Association between altered cognition and Loa loa microfilaremia: First evidence from a cross-sectional study in a rural area of the Republic of Congo" for consideration at PLOS Neglected Tropical Diseases. As with all papers reviewed by the journal, your manuscript was reviewed by members of the editorial board and by several independent reviewers. In light of the reviews (below this email), we would like to invite the resubmission of a significantly-revised version that takes into account the reviewers' comments. 

Please pay special attention to the comments from: Reviewer #3 regarding the use of MoCA testing for screening for cognitive impairment and need for subsequent confirmatory testing; Reviewer #2: additional information would be welcome regarding how cohort studies would help you address the limitations of this exploratory cross-sectional study; and Reviewer #1 regarding how test results were shared (or will be planned to be shared) with the public health system and population. 

We cannot make any decision about publication until we have seen the revised manuscript and your response to the reviewers' comments. Your revised manuscript is also likely to be sent to reviewers for further evaluation.

Sincerely,

Joseph Raymond Zunt

Academic Editor

Francesca Tamarozzi

Section Editor

Reviewer's Responses to Questions

**Key Review Criteria Required for Acceptance?**

**Methods**

-Are the objectives of the study clearly articulated with a clear testable hypothesis stated?

-Is the study design appropriate to address the stated objectives?

-Is the population clearly described and appropriate for the hypothesis being tested?

-Is the sample size sufficient to ensure adequate power to address the hypothesis being tested?

-Were correct statistical analysis used to support conclusions?

-Are there concerns about ethical or regulatory requirements being met?

Reviewer #1: The objectives are clearly articulated with clear testable hypothesis.

The study design is appropriate to address the stated objectives, but not clear specified as historical cohort study.

The population is clearly described and appropriate for the hypothesis being tested.

The sample size was not clearly calculted, but this is the forst time such study is conducted. It would heve been difficult to calculate the sample size without at least some preliminary results.

THe statistical analysis are appropriate

There is no concerns about ethical or regulatory requirement. They are all met.

Reviewer #2: Methodology: the objectives of the study are clearly articulated, the design is appropriate, given the study context, the population is well described, the sample size is sufficient, even though not justified, The statistical methods used are correct to support the conclusions. The study received ethical clearances from regulatory instances.

Reviewer #3: 1. The MOCA is a screening test that may over estimate the frequency of cognitive impairment. This is more likely when the cut-off is not based on local norms of the study area. The study should have used the 50 individuals with MF to get their mean score and then generate z scores to identify those who scores fall below 2 standard deviations of the control group mean score. 

2. The three groups are not well described in the methods. More details are needed.

**Results**

-Does the analysis presented match the analysis plan?

-Are the results clearly and completely presented?

-Are the figures (Tables, Images) of sufficient quality for clarity?

Reviewer #1: The results are clearly and completly presented. 

The tables are appropriate.

Reviewer #2: The analyses presented match the analysis plan and the results are clearly and completely presented. Tables and figures are of good quality and clarity.

Reviewer #3: 1. I find the frequency of impairment (>80%) too high to be true....and this is probably due how they determined impairment. A better method has been suggested above.

2. The results should be presented for the 3 different groups (high and low MF and the no MF group). How do the sociodemographic variables compared across the groups (Table 1) and MOCA scores (table 2).

**Conclusions**

-Are the conclusions supported by the data presented?

-Are the limitations of analysis clearly described?

-Do the authors discuss how these data can be helpful to advance our understanding of the topic under study?

-Is public health relevance addressed?

Reviewer #1: The conclusion are supported by the data presented. The limitation of the analysis are described but not complet. No recommendation was mad concerning further exploration in the domain despite the dougts concerning the results of the present study.

Reviewer #2: The conclusions are supported by the data presented. the limitations of the analysis are described, but more description need to be added to what is presented. The public health relevence is clearly addressed

Reviewer #3: 1. The discussion and conclusion should mention the limitations mentioned above (use of a screening test that may overestimate the impairment rates)

2. The authors in attempting to explain how MF lead to cognitive impairment could consider the BBB integrity being compromised by proinflammatory cytokines in response to infection. Future studies could consider investigating that.

**Editorial and Data Presentation Modifications?**

Reviewer #1: (No Response)

Reviewer #2: (No Response)

Reviewer #3: Data presentation modifications have been suggested above

**Summary and General Comments**

Reviewer #1: General comment

The paper by Checkouri et al on the association between altered cognition and Loa loa microfilaremia: first evidence from a cross-sectional study in a rural area of the Republic of Congo is an important contribution of the impact of loiasis, the neglected among the neglected by the scientific community. It is the first time the neuropsychiatric complications are described particularly in subject harboring very high MFD. In the logistic regression models and the Poisson regression models, there is no dose effect for the relation between the high MFD and the altered cognition statute. This put some doubt on this relation and as this has never been demonstrated elsewhere adding to possible confonding factors, the authors should clearly recommend the conduct of similar studies in different settings to confirm or to refute this relationship between the high loa MFD and the altered cognition.

Specific comment

Page 6

Study data were provided to community health workers in the involved villages to inform the local community of the study results and patients of their test results. I will suggest 

The results of the study were shared with the local health personnel to explain to the community members, and the individual results of the patients where put at their disposal through the local health personnel.

Page 8

The results of MoCA by a neurologist blinded to the parasitological results. It is not mentioned if the trained nurses examined the patients blinded of parasitological results.

Reviewer #2: The authors of this manuscript present for the first time an association between high density of Loa loa microfilariae in the peripheric blood and the alteration of cognition in a population living in endemic area of loiasis, independent of antifilarial treatment.

The findings from this pilote study constitute a breakthrough for this very neglected tropical disease (Loiasis) and deserve to be shared with scientific community working in this field of research.

The authors should highlight the fact that the study design they used “Cross-sectional” is the weakest design to establish causal relationship between exposure and disease in etiologic epidemiology. The good thing is that this study is nested in the cohort set up in a more ambitious project “Morbidity due to Loiasis” and one can expect that the findings from the long term follow up of participants would validate the preliminary observations.

Reviewer #3: The authors designed an exploratory study to examine the association between loa loa infection and cognitive functioning. Challenges observed are the use of a screening test and not using local norms to determine impairment which led to an unusually high rate of impairment.

PLOS authors have the option to publish the peer review history of their article (what does this mean?). If published, this will include your full peer review and any attached files.

Reviewer #1: Yes: Joseph Kamgno (MD, PhD)

Reviewer #2: No

Reviewer #3: No
---

## [Editor Report · Decision Letter 1]

3 Jun 2023

Dear Dr Campillo,

We are pleased to inform you that your manuscript 'Association between altered cognition and Loa loa microfilaremia: First evidence from a cross-sectional study in a rural area of the Republic of Congo' has been provisionally accepted for publication in PLOS Neglected Tropical Diseases.

Best regards,

Joseph Raymond Zunt

Academic Editor

Francesca Tamarozzi

Section Editor

Thank you for your thoughtful responses to the reviewers' comments. I do think it would be helpful to include the sensitivity analysis regarding MoCA score and L. loa MFD in the Supplementary material.

---

## [Editor Report · Acceptance letter]

14 Jun 2023

Dear Dr Campillo,

We are delighted to inform you that your manuscript, "Association between altered cognition and Loa loa microfilaremia: First evidence from a cross-sectional study in a rural area of the Republic of Congo," has been formally accepted for publication in PLOS Neglected Tropical Diseases.

Best regards,

Shaden Kamhawi

co-Editor-in-Chief

Paul Brindley

co-Editor-in-Chief
